# Maternal Education Level and Excessive Recreational Screen Time in Children: A Mediation Analysis

**DOI:** 10.3390/ijerph17238930

**Published:** 2020-12-01

**Authors:** Monserrat Pons, Miquel Bennasar-Veny, Aina M. Yañez

**Affiliations:** 1Department of Pediatrics, Hospital Son Espases, Carretera de Valldemossa, 79, 07120 Palma, Illes Balears, Spain; monserrat.pons@ssib.es; 2Department of Nursing and Physiotherapy, Balearic Islands University, Cra. de Valldemossa, km 7.5, 07122 Palma, Illes Balears, Spain; 3Research Group on Evidence, Lifestyles & Health, Health Research Institute of the Balearic Islands (IdISBa), 07010 Palma, Spain; aina.yanez@uib.es; 4Research Group on Global Health & Human Development, Balearic Islands University, Cra. de Valldemossa, km 7.5, 07122 Palma, Illes Balears, Spain

**Keywords:** screen-based media, children, structural equation modeling

## Abstract

There is increasing recognition of the adverse health consequences of excessive recreational screen time (RST) in children and adolescents. Early interventions that aim to reduce RST are crucial, but there are some controversies about which individual and parental variables affect RST in children. The aim of this study was to determine the relationship of parental education level with RST in children and early adolescents and to identify mediators of these relationships. This cross-sectional study examined a sample of children (2–14 year-old) who attended routine childcare visits in primary health care centers in Spain (*n* = 402; 53.7% males; mean age: 7 ± 4 year-old). A self-reported questionnaire was given to the parents to assess sociodemographic data, parental education, the home media environment, and RST in children. Separate analysis was performed for two age groups (2–6 year-old and 6–14 year-old). Path analysis, an application of structural equation modeling, was used to analyze the data. Fitty three percent of the children had excessive RST (≥2 h/day). The maternal education level, eating lunch/dinner in front of a TV, presence of a background TV, and the amount of parental TV viewing had significant associations with excessive RST in both age groups. For the younger group, the maternal education level had direct and indirect effects on RST (total effect: β = −0.29, *p* < 0.01). For the older group, maternal education level only had a significant indirect effect on RST, and this was mediated by the presence of a background TV and the time of parental TV viewing (total indirect effect: β = −0.11, *p* < 0.01). A higher maternal education level appears to be associated with certain environmental factors or habits that prevent excessive RST.

## 1. Introduction

Viewing of video screens is a very popular leisure-time sedentary behavior among children and adolescents, and includes watching television (TV), playing video games, using a computer, and surfing the web [1]. High percentages of European primary school children (6–9 year-old) exceed the internationally recommended limit of 2 h of recreational screen time (RST) per day; 19.0% to 31.7% exceed this limit on weekdays and 57.4% to 71.2% exceed this limit on weekends [2,3,4]. Studies of Spanish children showed similar results [5,6,7,8], but most studies examined children in narrow age groups [9]. Excessive RST is associated with several adverse health effects in young people. A systematic review of 235 studies from 71 different countries, in children and youth aged 5–17 years, reported positive associations of the duration of screen time with lower fitness and self-esteem, unfavorable body composition, and behavioral problems [10]. Furthermore, children with excessive RST tend to continue this behavior in adulthood [4,11].

Several studies indicated that watching TV and playing video games were inversely associated with the academic performance of children and adolescents [12,13]. However, other uses of screen media, such as surfing the web, using a computer, and using a mobile phone, had no such association, probably because these other types of screen media involve performing tasks, have more variable content, and often include non-recreational activities [14]. There is evidence that excessive television viewing by children is associated with inattentive/hyperactive behaviors, lower executive functions, and language delay, at least in the short-term [15]. Moreover, playing video games is associated with increased aggression, reduced prosocial behaviors, and symptoms of depression and attention deficit disorder [13]. Excessive playing of video games during childhood may also increase the risk for development of gaming disorder in adolescence [16].

In an attempt to reduce the negative effects of excessive RST, the Council on Communications and Media of the American Academy of Pediatrics recommended that parents should limit the non-educational screen time of their children to 2 h per day [2]. Recently, the World Health Organization (WHO) restricts the time of use of digital devices for children under 5 years of age to one hour or less [15]. In Spain, the Spanish Association of Primary Care Pediatrics recommends limiting the screen time to less than 2 h in general, without differentiating by age [16]. However, many European children and adolescents exceed this limit [8,17]. Early interventions that target excessive RST are crucial because screen viewing behaviors during adolescence are similar to those during childhood [18], and early adolescence is considered a critical period for the promotion of healthy behaviors [19].

It is important to identify the key correlates RST when designing interventions that attempt to reduce excessive RST. There are some controversial results about the associations of individual and parental variables with RST, but older age [6,20,21,22], parental TV time [6,20,21,22,23], presence of a TV in the bedroom [21,22,23], and male sex [1,21,24] were consistently reported as increasing the risk for excessive RST. There is also evidence of an association of parental education level with RST. Children from families with lower socioeconomic status have an increased risk for excessive RST, and excessive RST may exacerbate inequalities and lead to adverse health outcomes [6,24]. Few studies have evaluated the variables that mediate the relationship of parental education level and RST [25]. Identification and examination of the potential mediators of RST will help to develop tailored interventions that help less educated families to reduce RST and associated consequences in their children.

The main aim of this study was to examine the relationship of parental education level with RST in children and to evaluate factors in the home that mediate the associations between parental education and RST in children. We hypothesize that parents with less education engage in more sociocultural and environmental behaviors that increase the risk for excessive RST in their children.

## 2. Materials and Methods

### 2.1. Design and Sample

This cross-sectional study examined children who were 2 to 14 year-old and attended routine childcare visits in one of ten primary health care centers from the East Health Sector of the Balearic Islands (Spain) between January 2015 and June 2015. Exclusion criteria were moderate-to-severe pathology associated with significant physical limitations, hospital admission, or home rest of 2 or more weeks during the 3 months before enrolment.

Ethical approval was provided by the Research Ethics Committee of Manacor Hospital (Ref. 01/2014). The study was conducted according to the ethical guidelines of the Declaration of Helsinki. The parents of each participating family received an informational letter that explained the purpose of the study, and all parents provided signed written informed consent documents.

### 2.2. Measures

Pediatricians from selected health care centers invited to participate in the study parents who attended well-child visits during the study period. These visits are used to assess overall health, development and behavior and are programming at least once every year. Parents who accepted participate completed a written self-reported questionnaire (15–25 min) during the well-child visit. After 1 week 15 participants were asked to repeat the questionnaire to assess test-retest reliability.

The questionnaire collected sociodemographic data, home media environment, social and behavioral family characteristics related to TV viewing, parental TV time and children screen time. Sociodemographic data include age, gender, family structure, and parental education. Parental education indicated the highest educational level attained and was classified as: (a) incomplete primary education (less than 6 years); (b) complete primary education (6–8 years); (c) secondary compulsory education (>8–10 years); (d) upper-secondary education (>10–12 years); and (e) university degree. These 5 education categories were transformed into two categories: (a) secondary compulsory studies or lower; and (b) upper-secondary studies or higher. The primary and secondary school in Spain are compulsory for all children between 6 and 16 years (10 years).

The home media environment was assessed by three questions that asked about TV in the child’s bedroom (yes/no), number of TV sets at home, and number of other media devices at home (computer, tablet, game console, and other recreational children’s electronic devices with a screen). The total number of screens was summed to create a single number for video screens. The main sociocultural TV viewing habits were collected and rated using a 4 point Likert scale (0: always; 1: often; 2: occasionally; 3: never) for frequency of eating dinner and/or lunch in front of the TV, frequency of TV co-viewing with parents, frequency of TV viewing alone, and presence of a TV in the background. The four responses were categorized as often (0 and 1) and occasionally (2 and 3).

The times of TV viewing and playing video games were assessed separately for weekdays and weekends with 6 options (0–0.5 h, >0.5–1 h, >1–2 h, >2–3 h, >3–4 h, and >4 h). Parental TV time was recorded using the same options. Each interval was recoded as the midpoint of each class interval (class mark). Mean daily RST was obtained from the screen time of typical weekdays (multiplied by 5/7) and weekend days (multiplied by 2/7). Daily RST time was categorized as less than 2 h or more than 2 h.

### 2.3. Statistical Analysis

The test-retest reliability of the main items in the questionnaire was assessed after 1 week in 15 individuals. All the items had acceptable stability over time (kappa > 0.7) [26].

All analyses were performed separately for two age groups: preschool children (2–6 year-old) and school children/early adolescents (6–14 year-old). Descriptive statistics were used to characterize the subjects, and are expressed as mean ± SD for continuous variables and *n* (%) for categorical variables. To analyze the significance of differences between groups, Student’s *t*-test was used for continuous variables and the chi-square (χ^2^) test was used for categorical variables. 

A path analysis of structural equation modeling was used to analyze the data. The model was fitted multiple times using the maximum likelihood method. The fitting parameters included the chi-square test for goodness of fit (χ^2^/*df*), root means square error of approximation (RMSEA), comparative fit index (CFI), normal fit index, incremental fit index, and Tacker-Lewis Index (TLI) [27]. A model fit was considered acceptable if the χ^2^/*df* was below 5, the RMSEA was below 0.8, and the CFI and TLI were above 0.9. In this study, parental education level was used as an exogenous potential variable, and TV in the child’s bedroom, presence of a background TV, and parental TV time were used as endogenous potential variables. Age and sex were used as control variables.

All statistical analyses were performed using Statistical Package for Social Science (SPSS) version 25.0 and Amos version 23.0 (IBM Company, New York, NY, USA) in Microsoft Windows. All statistical tests were two-sided, and a *p*-value below 0.05 was considered significant.

## 3. Results

The parents of 432/504 children (85.7%) agreed to participate in this study. We excluded 30 participants who had missing data on screen time or parental education, so the final sample had 402 children (Table 1). Overall, 53% of the children had excessive RST (≥2 h/day), and this percentage was lower for children who were 2 to 6 year-old (50/132, 38%) than for those who were 6 to 14 year-old (162/270, 60%). Maternal education level, eating lunch or dinner in front of the TV, presence of a background TV, and parental TV watching were significantly associated with excessive RST in both age groups. Watching TV alone was associated with excessive RST in the older group and a TV in the child’s bedroom was associated with excessive RST in the younger group.

We also examined RST separately for TV and video games according to sex and age group (Figure 1). TV time and video game time increased with age in boys and girls. Boys spent more time playing video games than girls, and this difference was statistically significant for the older boys (*p* < 0.05).

The structural equation hypothetical models had good fits for the younger group (χ^2^/*df* = 8.175) and the older group (χ^2^/*df* = 6.2) and the fitting parameters indicated the measurement models were satisfactory (Figure 2 and Figure 3). For the older age group, the direct effect path for maternal education level on RST did not reach significance and was deleted (Figure 3). The hypothetical models were tested and corrected, and the final modified models had standardized path coefficients for the structural models. As hypothesized, for the younger group, maternal education level had a direct negative effect on RST (β = −0.18, *p* < 0.01). Also, maternal education level indirectly affected RST via TV in the child’s bedroom (β = −0.07, *p* < 0.01) and presence of a background TV (β = 0.05, *p* < 0.01). Children watching TV alone and parental TV time also affected RST (β = −0.18, *p* < 0.01). 

For the older group (Figure 3), maternal education level had an indirect effect on RST via the presence of a background TV (β = −0.03, *p* < 0.01) and parental TV time (β = 0.08, *p* < 0.01). Children watching TV alone (β = 0.17, *p* < 0.01) and sex (β = −0.14, *p* < 0.01) also affected RST.

## 4. Discussion

Our findings indicate that more than half of the children in our study population had excessive RST. Moreover, the mother’s education level was inversely associated with RST. This relationship was partly mediated by family socio-cultural practices and also differed according to child age. In the younger group (2–6 years old), TV in the child’s bedroom mediated this association while in the older group (>6–14) parental TV time was a mediator. Furthermore, background TV was a mediator variable in both groups. 

Previous studies also showed that maternal education, but not paternal education, was related to RST [28,29]. This difference could be because mothers tend to have a more important physical and emotional presence at the home [30,31,32]. Other longitudinal studies, with older population also reported that parental socioeconomic status was consistently associated with healthy lifestyles (tobacco consumption, quality of diet) during adolescence in Spain [32,33]. 

We found that parental TV time was strongly associated with RST in children. Previous studies attributed this effect to behavioral modeling by the children, although there is no evidence for this association for some other health-related behaviors, such as physical activity [5,34,35]. A possible explanation is that parental TV time provided an enabling environment for the children’s behavior. It is possible that parental TV time and presence of a background TV are related, and could also function as a cue to action [36], thus influencing children’s behavior [4].

There is little known about the association between parental education level and the presence of a background TV. As we hypothesized, mothers with less education were more likely to have a background TV, and this variable mediated the association between parental education and RST in children. Similarly, we also found that more than 90% of the homes that had a background TV also had the TV on during meals (data not shown). Therefore, we did not include this variable in the structural equation model, because we considered the presence of a background TV to include having a TV on during meals. A previous study that evaluates a similar age range children [37] and other with pre-school children [38] also reported an association between having a TV on during meals and RST. It is important to highlight previous research which reported that a background TV and a TV on during meals were related to fewer parent-child interactions, and these interactions are crucial to promoting the mental development of children [39]. Other factors related to socioeconomic status could also explain the excessive RST of children from families whose mothers had less education. Some barriers to limiting RST in children were reported previously, such as lacking an affordable alternative entertainment, the need to keep children occupied while performing chores, parental exhaustion or fatigue, and desire for time away from children to complete other activities [40]. All these barriers could be especially important in families with lower socioeconomic status.

Although our sample was mostly Spanish (96%), in the bivariate analysis we observed that nationality (other versus Spanish) was significantly associated with excessive recreational screen time in younger children. These findings are in line with other studies [41] and with one systematic review, where the evidence showed that screen time was consistently found to be higher in non-Caucasian compared to Caucasian participants [20].

We also compared the relationship of different variables with RST in children from two age groups. For young children, maternal education level was directly associated with RST, and partially mediated by a TV in the child’s bedroom and the presence of a background TV. However, for older children, maternal education level only had an indirect effect on RST, and this was mediated by the presence of a background TV and parental TV time. These differences, which were reported previously, could be explained by the greater autonomy of older children [42]. Moreover, a TV in a child’s bedroom seemed to be related to excessive RST only in younger children. In agreement with our results, Saelens et al. [43] reported age differences in the associations of different factors with time watching TV, in that younger children watched more TV when they had a TV in their bedrooms but there was no such association for older children. Other studies of preadolescents and adolescents also reported no association between the presence of a TV in the bedroom and RST in older children [7,44]. It is possible that the greater autonomy of older children allows them to view recreational screens without being detected by their parents, as reported by Robinson et al. [45].

In addition to the environmental and sociocultural variables examined here, individual differences also affect the viewing of recreational screens. Our results indicated that watching TV alone is associated with excessive RST in both age groups; there is limited study of this factor in the literature. However, co-viewing of TV with parents was reported to be positively associated with RST and inversely associated with parental education level [37,46,47]. Similar to our results, a recent study found no significant associations of co-viewing TV with parental education level or with RST [48]. This may indicate that although co-viewing provides parents with an opportunity to monitor and accompany their children, it may not have a strong influence on the child’s RST. Instead, other practices, such as frequently watching TV alone, appear to indicate a strong effect on RST and are probably most relevant when attempting to limit excessive RST in children and adolescents.

We found that boys played more video games than girls and that this difference was statistically significant for the older age group. Our path analysis also indicated an association between sex and RST only in the older group, probably due to the increased playing of video games by boys [42]. Notably, excessive video gaming is a global public health concern that needs to be addressed using a more integrative approach [49]. This addiction can have severe repercussions, especially during adolescence [50]. Although we have not identified all the factors that affect video gaming, there is evidence that more video gaming is a risk factor for subsequent video game addiction [16].

Our findings indicated no significant relationship between the number of screens in the home and RST, in contrast to a previous report [51]. Our results showed that RST and the number of screens increased with age, but separate analysis of the two age groups indicated the number of screens in the home did not impact RST.

Our study had several limitations. We used indirect measures to record data on RST by questioning the parents. Despite the potential for desirability bias and the possible difficulties parents may have in accurately reporting RST in their children, use of questionnaires allowed us to achieve a more representative sample at less cost than other methods, such as daily diaries [52]. The high level of consistency in our test and re-test measurement of reliability support the validity of our questionnaire. Furthermore, most other studies that examined RST also used indirect methods for measurement of screen time.

Although we used a structural equation model, our study had a cross-sectional design. Thus, we cannot establish causal relationships of different variables with RST. But, some of the evaluated variables, such as parental education are previous to the children’s behavior. Finally, our data were collected during 2015 and some changes could have happened, particularly in the amount of time that children spend using recreational screens. In any case, it is likely that the risk factors for excessive use remain longer over time.

The strengths of the present study are that we examined children aged 2 to 14 year-old and separately examined two age groups. Most studies of RST examined children in narrower age ranges. Therefore, the results of our study provide data about differences in risk factors for children of different ages.

This study provides insights into strategies that may be used to reduce the excessive RST in children and young adults. In particular, our results point to the importance of reducing parental TV time, turn off the TV when no one is watching, not turning on the TV during meals and not placing TV in the children’s bedroom as the major factors to prevent excessive RST.

## 5. Conclusions

The present study found that more than half of the children had excessive RST, and that excessive RST was more common in the older group. Maternal education is a key factor affecting the child’s RST. Parents, especially mothers, have a strong influence in the home environment and the mother’s education level can affect certain environmental factors (e.g., TV in the child’s bedroom) and habits (e.g., background TV) that influence RST in the child. Further research about the effectiveness of interventions to reduce RST, that include the framework of social determinants of health in the intervention design could contribute to reduce the perpetuation of health inequities between generations. Our results suggest that health professionals and educators should consider the social characteristics of families when giving advice on TV use. Parents who want to reduce RST in their child should focus on several key variables: the child watching TV alone, a TV in the child’s bedroom, presence of a background TV, and duration of parental TV time.

## Figures and Tables

**Figure 1 ijerph-17-08930-f001:**
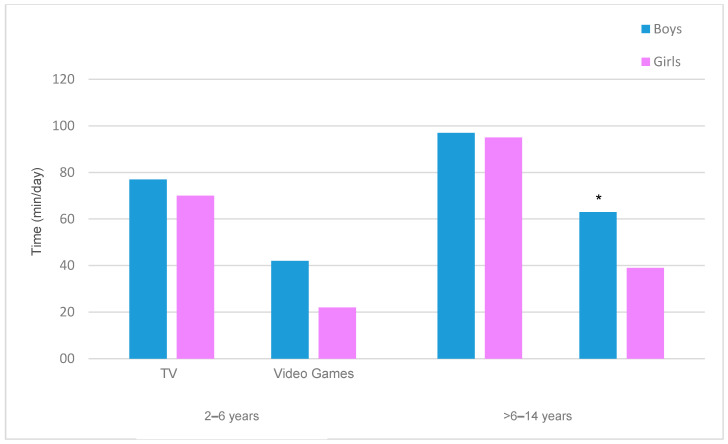
Time watching TV and playing video games by boys and girls in the two age groups. * Statistically significant differences (*p* < 0.05) for the comparison of video games between boys and girls.

**Figure 2 ijerph-17-08930-f002:**
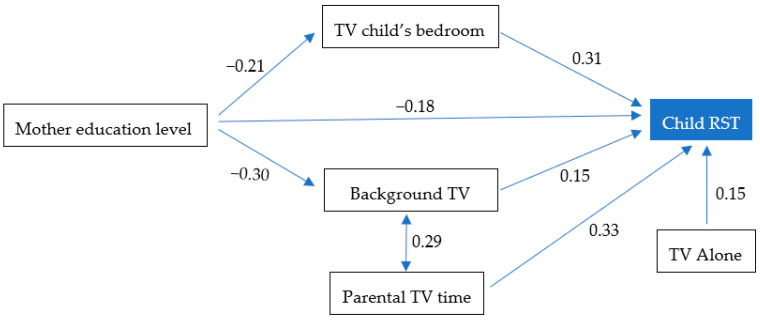
Final Path Model for the younger group (2–6 year-old) with standardized regression coefficients and correlations between variables. Proportion of variance explained: 38%. Fitting parameters: χ^2^ = 8.175, *df* = 6, *p* < 0.01; RMSEA < 0.01; CFI = 1; TLI = 0.95. TV, television; RST, recreational screen time; RMSEA, root means square error of approximation; CFI, comparative fit index; TLI, Tacker-Lewis index.

**Figure 3 ijerph-17-08930-f003:**
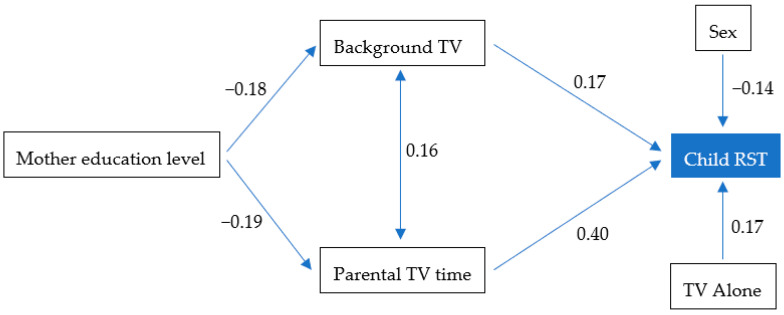
Final Path Model for older group (6–14 year-old) with standardized regression coefficients and correlations between variables. Proportion of variance explained: 27%. Fitting parameters: χ^2^ = 6.197, *df* = 6, *p* < 0.01; RMSEA < 0.01; CFI = 1; TLI = 1.014. TV, television; RST, recreational screen time; RMSEA, root means square error of approximation; CFI, comparative fit index; TLI, Tacker-Lewis index.

**Table 1 ijerph-17-08930-t001:** Association of different variables with recreational screen time (more or less than 2 h) in two age groups of children.

Daily Screen Time							
	Total	2–6 Years			>6–14 Years		
Variable	N	<2 h/day	≥2 h/day	*p*-Value	<2 h/day	≥2 h/day	*p*-Value
Sex				0.054			**0.004**
Boy	216	35 (53.8)	30 (46.2)		49 (32.5)	102 (67.5)	
Girl	186	47 (70.1)	20 (29.9)		59 (49.6)	60 (50.4)	
Type of family				0.463			0.124
Traditional	320	70 (63.6)	40 (36.4)		90 (42.9)	120 (57.1)	
Single parent or other	77	11 (55.0)	9 (45.0)		18 (31.6)	39 (68.4)	
Nationality				**0.030**			0.681
Spanish	379	80 (60.4)	45 (36.0)		102 (40.2)	152 (59.8)	
Foreign	15	1 (16.7)	5 (83.3)		3 (33.3)	6 (66.7)	
Father education level				0.098			0.722
Primary or less	221	44 (61.1)	28 (38.9)		61 (40.9)	88 (59.1)	
Secondary or more	132	37 (75.5)	12 (34.5)		32 (38.6)	51 (61.4)	
Mother education level				**<0.001**			**0.030**
Primary or less	187	22 (42.3)	30 (57.7)		45 (33.3)	90 (66.7)	
Secondary or more	203	57 (75.0)	19 (25.0)		59 (46.5)	68 (53.5)	
Lunch/dinner in front of TV				**0.008**			**0.024**
Occasionally	119	33 (78.6)	9 (24.4)		39 (50.6)	38 (49.4)	
Often	283	49 (54.4)	41 (45.6)		69 (35.8)	124 (64.2)	
Viewing TV alone				0.134			**0.001**
Occasionally	147	40 (69.0)	18 (31.0)		48 (53.9)	41 (46.1)	
Often	252	41 (56.2)	32 (43.8)		59 (33.0)	120 (67.0)	
Co-viewing TV				0.095			0.129
Occasionally	71	13 (81.3)	3 (18.8)		27 (49.1)	28 (50.9)	
Often	328	68 (59.6)	46 (40.4)		81 (37.9)	133 (62.1)	
Background TV				**0.006**			**<0.001**
Occasionally	211	45 (73.8)	16 (26.2)		75 (50.0)	75 (50%)	
Often	129	34 (50.0)	34 (50.0)		32 (27.8)	83 (72.2)	
TV in child’s bedroom				**<0.001**			0.227
No	270	73 (70.9)	30 (29.1)		72 (43.1)	95 (56.9)	
Yes	129	8 (28.6)	20 (71.4)		36 (35.6)	65 (64.4)	
Number of screens in house				0.102			0.149
≤5	240	66 (67.3)	32 (32.7)		63 (44.4)	79 (55.6)	
≥6	131	13 (50%)	13 (50%)		37 (35.2)	68 (64.8)	
Parental TV time		73.0 ± 43.5	104.0 ± 53.3	**0.001** **0.004**	61.7 ± 41.3	105.9 ±53.4	**<0.001** **<0.001**
<2 h/day	296	68 (67.3)	33 (32.7)		97 (49.7)	98 (50.3)	
≥2 h/day	92	9 (36.0)	16 (64.0)		9 (13.4)	58 (86.6)	

Bold values denote statistical significance at the *p*-value < 0.05 level.

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
