# Peer review of "Maternal Education Level and Excessive Recreational Screen Time in Children: A Mediation Analysis"

_ijerph, 2020, doi:10.3390/ijerph17238930_

Round 1
Reviewer 1 Report
Methods
- My suggestion is to unify paragraphs 2.1 and 2.4. Moreover, I would enrich the design description by explaining better the adopted administration and sampling strategies and, if available, how the questionnaire was validated.
- What is intended by self-administered questionnaire (line 87 – page 2)? Do you mean self-reported? As mentioned in the previous comment, I would encourage you to better describe the questionnaire administration strategy.
- At the beginning of the “measures” paragraph, I would include an introductory sentence to specify all the dimensions investigated in the questionnaire, before describing more specifically the data collected. If possible, maybe you could include the questionnaire in an appendix.
- First, I would describe what data you collected, and then how you included such measures in the statistical analysis. The following sentences: “These 5 education categories were transformed into two categories: (a) secondary compulsory studies or lower; and (b) upper-secondary studies or higher.” (lines 92-93 - page 2), “The total number of screens was summed to create a single number for video screens.” (line 97 – page 3), “The four responses were categorized as often (0 and 1) and occasionally (2 and 3).” (line 101 – page 3) and “Daily RST time was categorized as less than 2 h or more 106 than 2 h.” (lines 106-107) would better fit in a description of how you included the variables in the analysis.
Discussion
- I think that the results of your study are very interesting. In particular, regarding this sentence: This study provides insights into strategies that may be used to reduce the excessive RST in children and young adults (lines 252-253 – page 7), could you please try to explode a bit what you have in mind as for these strategies in order to make your findings more actionable?
Conclusions
- Please, consider revising “RCT” (lines 260 and 261 - page 8).
- I would recommend that you present this (lines 261-263 - page 8) in terms of opportunities for further research.
- It is more or less the same what I tried to suggest in the discussion section. I believe that your findings are very interesting and, in particular, it is important that they can be relevant for parents, but how do they inform policy-makers and health professionals? How can these categories use your results in practical terms in order to improve children’s health status effectively?
Author Response
Dear Editor and reviewers,
We would like to thank the Editor and reviewers the time dedicated to revising the manuscript, their assessments and all the comments made on this work. We agree with most of the observations and are convinced that have improved the clarity and scientific value of this research. We apologize for the mistakes, and we have also corrected the text as suggested. Track changes from Microsoft Word show the modifications performed by the authors. To make the changes easier to understand, we have including, when possible, the revised added texts in the manuscript as answers to your comments.
Thank you very much for your time and interest.
We look forward to hearing from you soon.
Sincerely,
Miquel Bennasar-Veny
Response to Reviewer 1 Comments
Methods
My suggestion is to unify paragraphs 2.1 and 2.4. Moreover, I would enrich the design description by explaining better the adopted administration and sampling strategies and, if available, how the questionnaire was validated.
R: We appreciate the recommendation, thank you. This has been changed according to the reviewer comments. Also, we have added more information about the adopted administration and sampling strategies (page 3).
As we mention in the manuscript, we have evaluated the reliability of the questionnaire.
What is intended by self-administered questionnaire (line 87 – page 2)? Do you mean, self-reported? As mentioned in the previous comment, I would encourage you to better describe the questionnaire administration strategy.
R: We mean self-administered, and we have changed the text according to the reviewer suggestion.
At the beginning of the “measures” paragraph, I would include an introductory sentence to specify all the dimensions investigated in the questionnaire, before describing more specifically the data collected. If possible, maybe you could include the questionnaire in an appendix.
R: Thank you for your suggestion. An introductory sentence was added to specify questionnaire dimensions and administration strategy was also explained in more detail: “Pediatricians from selected health care centers invited to participate in the study parents who attended well-child visits during the study period. These visits are used to assess overall health, development and behavior and are programming at least once every year. Parents who accepted participate completed a written self-reported questionnaire (15–25 min) during the well-child visit. The questionnaire collected sociodemographic data, home media environment, social and behavioral family characteristics related to TV viewing, parental TV time and children screen time. Sociodemographic data include age, gender, family structure, and parental education during routine childcare visits”.
First, I would describe what data you collected, and then how you included such measures in the statistical analysis. The following sentences: “These 5 education categories were transformed into two categories: (a) secondary compulsory studies or lower; and (b) upper-secondary studies or higher.” (lines 92-93 - page 2), “The total number of screens was summed to create a single number for video screens.” (line 97 – page 3), “The four responses were categorized as often (0 and 1) and occasionally (2 and 3).” (line 101 – page 3) and “Daily RST time was categorized as less than 2 h or more 106 than 2 h.” (lines 106-107) would better fit in a description of how you included the variables in the analysis.
R: Thank you for your suggestion. As we have now included an introductory sentence to describe the dimensions of the questionnaire, we finally decided to left categorization descriptions next to the description of data collection.
Discussion
I think that the results of your study are very interesting. In particular, regarding this sentence: This study provides insights into strategies that may be used to reduce the excessive RST in children and young adults (lines 252-253 – page 7), could you please try to explode a bit what you have in mind as for these strategies in order to make your findings more actionable?
R: Following your suggestion, we have changed this sentence in terms of actionable findings.
Page 8, Line 287-290: “In particular, our results point to the importance of reducing parental TV time, turn off the TV when no one is watching, not turning on the TV during meals and not placing the TV in the children's bedroom as the major factors to prevent excessive RST”.
Conclusions
Please, consider revising “RCT” (lines 260 and 261 - page 8).
R: We are sorry by this mistake. This has been amended.
I would recommend that you present this (lines 261-263 - page 8) in terms of opportunities for further research.
R: This has been modified according to the reviewer comments.
Page 8, Line 299-301: “Further research about the effectiveness of interventions to reduce RST, that include the framework of social determinants of health in the intervention design could contribute to reduce the perpetuation of health inequities between generations”.
It is more or less the same what I tried to suggest in the discussion section. I believe that your findings are very interesting and, in particular, it is important that they can be relevant for parents, but how do they inform policy-makers and health professionals? How can these categories use your results in practical terms in order to improve children’s health status effectively?
R: Thank you for this insightful observation. We have added some recommendations for health educators and health professional according to our results.
Page 8, Line 302-304: “Our results suggest that health professionals and educators should consider the social characteristics of families when advising on TV use. Parents …”

Reviewer 2 Report
Dear authors,
I have completed the review or your manuscript IJERPH 997644.
Please find below my comments.
Best Regards.
IJERPH 997644
The study submitted by Pons et alia describes how maternal education level influence recreational screen time (RST) in children aged 2-6 and 6-14 years old. They also outline several other variables as being linked to RST.
The manuscript is well written, easy to understand and adequately structured. Conclusions are clearly supported by results.
However, a few alterations are needed before publication. These are outlined below.
1/ General comment. Having a TV screen in the child’s bedroom does not necessarily mean that it is used (ON). Do authors have any data regarding the use of the bedroom’s TV by the children concerned ?
2/ General comment. Throughout the manuscript, authors used a cut-off criterion at 2 hours/day. My understanding is that it is from a National Guideline. However, 2 hours per day seems very high for children aged 2 years old, while is seems adequate for children aged 14 years old. Can the authors put those values into perspective ? What about guidelines according to age AND country ?
3/ Lines 49-52, authors rightly mention that some screen time (as opposed to TV time) can be beneficial, especially for school (i.e. education). This is very interesting. Do author have any data on such screen time ? This should also be mentioned, again, in the discussion section.
4/ Methods line 83. Data was acquired between January 2015 and June 2015. It has now been several years since data collection. Could author provide any recent data ? If not, the discussion should also reflect the fact that changes could have happened since 2015, for the good or worse…
5/ Methods line 89-93. The level of education is classified according to years spent in the schooling system. While this is perfectly fine, could authors provide a sentence detailing how many years of schooling are mandatory in Spain (or more locally, i.e. the Balearic Islands). This is to put their data into perspective.
6/ Legend of Table 1. My understanding is that bold statistics reflect significant values. Please insert such a caption.
7/ Figure 1 is drawn with straight lines between the two groups for each variable (gender, TV, games). However, this suggests paired variables (i.e. TV screen time in boys measured when the subjects were 2-6 years old then measured again at 6-14 years old). Please correct this Figure to reflect that the two measures (2-6 vs 6-14) are not related.
8/ Line 179, “could because”. One verb is missing. Please correct.
9/ Line 180, should it rather read “at home”, instead of “in the home” ?
10/ General comment. Table 1 also presents “nationality” as a significant factor. Could the author comment such a result ? Could authors provide a small discussion on this variable ?
11/ General comment. Figures 2 and 3 present hypothetical models. Please specifically mention somewhere in your manuscript that it is hypothetical.
Author Response
Dear Editor and reviewers,
We would like to thank the Editor and reviewers the time dedicated to revising the manuscript, their assessments and all the comments made on this work. We agree with most of the observations and are convinced that have improved the clarity and scientific value of this research. We apologize for the mistakes, and we have also corrected the text as suggested. Track changes from Microsoft Word show the modifications performed by the authors. To make the changes easier to understand, we have including, when possible, the revised added texts in the manuscript as answers to your comments.
Thank you very much for your time and interest.
We look forward to hearing from you soon.
Sincerely,
Miquel Bennasar-Veny
Response to Reviewer 2 Comments
The study submitted by Pons et alia describes how maternal education level influence recreational screen time (RST) in children aged 2-6 and 6-14 years old. They also outline several other variables as being linked to RST.
The manuscript is well written, easy to understand and adequately structured. Conclusions are clearly supported by results.
However, a few alterations are needed before publication. These are outlined below.
R: We are very grateful to the reviewer for their encouragement.
1/ General comment. Having a TV screen in the child’s bedroom does not necessarily mean that it is used (ON). Do authors have any data regarding the use of the bedroom’s TV by the children concerned?
R: We agree with the reviewer that it could be very interesting, but unfortunately, we do not have data about TV use in the child’s bedroom.
2/ General comment. Throughout the manuscript, authors used a cut-off criterion at 2 hours/day. My understanding is that it is from a National Guideline. However, 2 hours per day seems very high for children aged 2 years old, while is seems adequate for children aged 14 years old. Can the authors put those values into perspective? What about guidelines according to age and country?
R: We appreciate this comment, thank you. Following your suggestion, we have added some information in the manuscript about different cut-off criteria for RST according to age and country.
Lines 61-64: “Recently, the World Health Organization (WHO) restricts the time of use of digital devices for children under 5 years of age to one hour or less [15]. In Spain, the Spanish Association of Primary Care Pediatrics recommends limiting the screen time to less than 2 hours in general, without differentiating by age [16]”.
3/ Lines 49-52, authors rightly mention that some screen time (as opposed to TV time) can be beneficial, especially for school (i.e. education). This is very interesting. Do author have any data on such screen time? This should also be mentioned, again, in the discussion section.
R: We are sorry, but we do not have data about non-recreational screen time.
4/ Methods line 83. Data was acquired between January 2015 and June 2015. It has now been several years since data collection. Could author provide any recent data? If not, the discussion should also reflect the fact that changes could have happened since 2015, for the good or worse…
R: Thank you for the suggestion. We have now reflected your suggestions in the discussion section (page 8, lines 278-281): “Finally, our data were collected during 2015 and some changes could have happened, particularly in the amount of time that children spend using recreational screens. In any case, it is likely that the risk factors for excessive use remain longer over time”.
5/ Methods line 89-93. The level of education is classified according to years spent in the schooling system. While this is perfectly fine, could authors provide a sentence detailing how many years of schooling are mandatory in Spain (or more locally, i.e. the Balearic Islands). This is to put their data into perspective.
R: We are agreeing with the reviewer and we include this information in the methods section (lines 109-110): “The primary and secondary school in Spain are compulsory for all children between 6 and 16 years (10 years)”.
6/ Legend of Table 1. My understanding is that bold statistics reflect significant values. Please insert such a caption.
R: We are grateful to the reviewer for this suggestion. This has been modified. Now, the legend of Table 1 includes “Bold values denote statistical significance at the p-value < 0.05 level”.
7/ Figure 1 is drawn with straight lines between the two groups for each variable (gender, TV, games). However, this suggests paired variables (i.e. TV screen time in boys measured when the subjects were 2-6 years old then measured again at 6-14 years old). Please correct this Figure to reflect that the two measures (2-6 vs 6-14) are not related.
R: Thank you for your suggestions. We have changed the Figure 1 by a bar chart to reflect that these two measures are not related.
8/ Line 179, “could because”. One verb is missing. Please correct.
R: Sorry for this mistake. This has been changed.
9/ Line 180, should it rather read “at home”, instead of “in the home”?
R: This has been changed.
10/ General comment. Table 1 also presents “nationality” as a significant factor. Could the author comment such a result? Could authors provide a small discussion on this variable?
R: Following your recommendation, we have added a small discussion on this variable.
Page 7, Lines 230-234: “Although our sample was mostly Spanish (96%), in the bivariate analysis we observed that nationality (other versus Spanish) was significantly associated with excessive recreational screen time in younger children. These findings are in line with other studies [42] and with one systematic review, where the evidence showed that screen time was consistently found to be higher in non-Caucasian compared to Caucasian participants [20]”.
11/ General comment. Figures 2 and 3 present hypothetical models. Please specifically mention somewhere in your manuscript that it is hypothetical.
R: Thank you for the suggestion. We have added this information in the results section (line 171 and line 174).

Reviewer 3 Report
- Thank you for the opportunity to review this manuscript. Overall, this is an interesting study investigating the relationship of parental education level with RST in children and early adolescents and to identify mediators of these relationships.
General suggestions:
- This article would benefit from using the Strengthening the Reporting of Observational Studies in Epidemiology (STROBE).
- The quality of the article would benefit from having a professional Academic English teacher to review the manuscript. For example, in Page 6 - line 179 “This difference could [ ] because…”; Page 7 - lines 206-207 “However, for older children maternal education level only had an indirect [ ] on RST.
I provide major comments below for the authors' consideration:
Introduction:
- Page 1, line 39-40: It would be useful to add the current WHO movement guidelines for children under 5 years old, emphasizing screen time behaviour.
- Page 1-2, lines 44-46: “A systematic review of 235 studies from 71 different countries reported positive associations…” – Please specify which age group were targeted.
- Page 2, line 48: “Several studies indicated that watching TV and playing video games were inversely associated with the academic performance of children and adolescents”. – This sentence needs citation.
- Page 2, lines 59-61: “… the Council on Communications and Media of the American Academy of Pediatrics recommended that parents should limit the non educational screen time of their children to 2 h per day [2]. However, many children and adolescents exceed this limit [8, 15]. From which countries children are exceeding the guidelines for screen time, and in which age group? – Please be more specific.
Methods:
- Page 2, lines 81-82: “This cross-sectional study examined children who were 2 to 14 years-old and attended routine childcare visits in one of ten primary health care centers from the East Health Sector of the Balearic Islands (Spain) - How the Child Care center was selected? Was it randomly selected? Was there any sample size calculation?
- Page 3, lines 87-107: Could the authors please provide some more information about the questionnaire used? From which sources they were obtained? It would be important to include the psychometric properties of the tool, and/or if it has been used in previous studies.
- Page 3, lines 109-110: “The test-retest reliability of the main items in the questionnaire was assessed after 1 week in 15 individuals. All the items had acceptable stability over time (kappa > 0.7)” – Why only in 15 individuals? Was it enough? This information should be placed in the “Measures” section.
- Page 3, lines 117-119: “The fitting parameters included the chi square test for goodness of fit (χ2/df), root means square error of approximation (RMSEA)…” – This sentence requires a citation.
Results:
- The results are clear and well descried.
Discussion:
- Page 8-9, lines 284-292 – I suggest adding in the first paragraph the main results of the study, and then discussing the findings more precisely in the light of the existing evidence.
- In general, there is a need to discuss the findings of other studies in more detail. For example “Other studies also reported that parental socioeconomic status was consistently associated with healthy lifestyles during childhood and adolescence [30, 31]” – Which health lifestyles are they? Please describe the studies in more detail… “Other studies also reported an association between having a TV on during meals and RST [35, 36]”… Please provide details of these studies, including the targeted age group, population and how data in these studies were collected.
- The study would benefit from providing the implications in terms of targets for future studies/ interventions.
Conclusions:
16. Page 8, lines 259-262: “Parents, especially mothers, have a strong influence in the home environment and the mother’s education level can affect certain environmental factors and habits that influence RCT in the child” – Which environmental factors and habits are they? Please, report.
Author Response
Dear Editor and reviewers,
We would like to thank the Editor and reviewers the time dedicated to revising the manuscript, their assessments and all the comments made on this work. We agree with most of the observations and are convinced that have improved the clarity and scientific value of this research. We apologize for the mistakes, and we have also corrected the text as suggested. Track changes from Microsoft Word show the modifications performed by the authors. To make the changes easier to understand, we have including, when possible, the revised added texts in the manuscript as answers to your comments.
Thank you very much for your time and interest.
We look forward to hearing from you soon.
Sincerely,
Miquel Bennasar-Veny
Response to Reviewer 3 Comments
- Thank you for the opportunity to review this manuscript. Overall, this is an interesting study investigating the relationship of parental education level with RST in children and early adolescents and to identify mediators of these relationships.
R: We would like to thank the reviewer for the careful and thorough reading of this manuscript.
General suggestions:
- This article would benefit from using theStrengthening the Reporting of Observational Studies in Epidemiology (STROBE).
R: Thank you for your suggestion. We have added some information regarding point 6 of STROBE checklist (Participants: Give the eligibility criteria, and the sources and methods of selection of participants) at the methods section.
- The quality of the article would benefit from having a professional Academic English teacher to review the manuscript. For example, in Page 6 - line 179 “This difference could [ ] because…”; Page 7 - lines 206-207 “However, for older children maternal education level only had an indirect [ ] on RST.
R: We appreciate the recommendation, thank you. We have reviewed all the manuscript for English spelling.
I provide major comments below for the authors' consideration:
Introduction:
- Page 1, line 39-40:It would be useful to add the current WHO movement guidelines for children under 5 years old, emphasizing screen time behaviour.
R: We are grateful to the reviewer for this recommendation. We have added this information in the introduction section (Page 2, lines 61-64), and now reads: “Recently, the World Health Organization (WHO) restricts the time of use of digital devices for children under 5 years of age to one hour or less [15]. In Spain, the Spanish Association of Primary Care Pediatrics recommends limiting the screen time to less than 2 hours in general, without differentiating by age [16]”.
- Page 1-2, lines 44-46:“A systematic review of 235 studies from 71 different countries reported positive associations…” – Please specify which age group were targeted.
R: Thank you for this suggestion, we have added this information (page 1, line 45).
- Page 2, line 48:“Several studies indicated that watching TV and playing video games were inversely associated with the academic performance of children and adolescents”. – This sentence needs citation.
R: We are grateful for the suggestion and have now added two references for this sentence:
- Tremblay, M. S.; et al., Systematic review of sedentary behaviour and health indicators in school-aged children and youth. Int J Behav Nutr Phys Act 2011, 8, 98.
- Ferguson, C. J. Do Angry Birds Make for Angry Children? A Meta-Analysis of Video Game Influences on Children's and Adolescents' Aggression, Mental Health, Prosocial Behavior, and Academic Performance. Perspect Psychol Sci 2015, 10, (5), 646-66.
- Page 2, lines 59-61: “…the Council on Communications and Media of the American Academy of Pediatrics recommended that parents should limit the non educational screen time of their children to 2 h per day [2]. However, many children and adolescents exceed this limit [8, 15]. From which countries children are exceeding the guidelines for screen time, and in which age group? – Please be more specific.
R: Many thanks for this clarification, we have added this information in the manuscript: “However, many European children and adolescents…” (page 2, line 64).
Methods:
- Page 2, lines 81-82: “This cross-sectional study examined children who were 2 to 14 years-old and attended routine childcare visits in one of ten primary health care centers from the East Health Sector of the Balearic Islands (Spain) - How the Child Care center was selected? Was it randomly selected? Was there any sample size calculation?
R: We have added some information about sample selection. Pediatricians offered all parents who attended well-child visit during study period to participate in the study.
Page 3, lines 95-103: “Pediatricians from selected health care centers invited to participate in the study parents who attended well-child visits during the study period. These visits are used to assess overall health, development and behavior and are programming at least once every year. Parents who accepted participate completed a written self-reported questionnaire (15–25 min) during the well-child visit. The questionnaire collected sociodemographic data, home media environment, social and behavioral family characteristics related to TV viewing, parental TV time and children screen time. Sociodemographic data include age, gender, family structure, and parental education”.
Regarding the sample size, we had calculated that a sample size of 385 children will suffice to estimate with a 95% confidence and a precision +/- 5 percent units considering the percentage around 50%.
- Page 3, lines 87-107:Could the authors please provide some more information about the questionnaire used? From which sources they were obtained? It would be important to include the psychometric properties of the tool, and/or if it has been used in previous studies.
R: We have developed our questionnaire specifically for the study. As we mention in the manuscript, we perform test-retest reliability of the main items in the questionnaire, and all the items had acceptable stability over time.
- Page 3, lines 109-110: “The test-retest reliability of the main items in the questionnaire was assessed after 1 week in 15 individuals. All the items had acceptable stability over time (kappa > 0.7)” – Why only in 15 individuals? Was it enough? This information should be placed in the “Measures” section.
R: We performed the test-retest reliability with only 15 individuals because we have limited time and financial resources. Thank you for the suggestion; we have placed this information in the measures section (page 3, lines 99-100).
- Page 3, lines 117-119: “The fitting parameters included the chi square test for goodness of fit (χ2/df), root means square error of approximation (RMSEA)…” – This sentence requires a citation.
R: We are grateful for the suggestion and have now added a reference for this sentence:
- Hu, L. T., & Bentler, P. M. (1999). Cutoff criteria for fit indexes in covariance structure analysis: Conventional criteria versus new alternatives. Structural Equation Modeling, 6(1), 1–55.
Results:
- The results are clear and well described.
R: Thank you ever so much for your comments.
Discussion:
- Page 8-9, lines 284-292– I suggest adding in the first paragraph the main results of the study, and then discussing the findings more precisely in the light of the existing evidence.
R: Thank you for your suggestion. We have added the main results in the first paragraph: “In the younger group (2-6 years old), TV in the child’s bedroom mediated this association while in the older group (>6-14) parental TV time was a mediator. Furthermore, background TV was a mediator variable in both groups”.
- In general, there is a need to discuss the findings of other studies in more detail. For example, “Other studies also reported that parental socioeconomic status was consistently associated with healthy lifestyles during childhood and adolescence [30, 31]” – Which health lifestyles are they? Please describe the studies in more detail… “Other studies also reported an association between having a TV on during meals and RST [35, 36]”… Please provide details of these studies, including the targeted age group, population and how data in these studies were collected.
R: We have added some information about the mentioned studies.
Page 7, Lines 204-206: “Other longitudinal studies, with older population, also reported that parental socioeconomic status was consistently associated with healthy lifestyles (tobacco consumption, quality of diet) during childhood and adolescence in Spain [33, 34]”.
Page 7, Lines 219-221: “A previous study that evaluates a similar age range children [38] and other with pre-school children [39] also reported an association between having a TV on during meals and RST”.
- The study would benefit from providing the implications in terms of targets for future studies/ interventions.
R: Thank you for the suggestion. We have rewritten some sentence in terms of opportunities for future research.
Page 8, Lines 299-302: “Further research about the effectiveness of interventions to reduce RST, that include the framework of social determinants of health in the intervention design could contribute to reduce the perpetuation of health inequities between generations”.
Conclusions:
- Page 8, lines 259-262: “Parents, especially mothers, have a strong influence in the home environment and the mother’s education level can affect certain environmental factors and habits that influence RST in the child” – Which environmental factors and habits are they? Please, report.
R: Following your suggestion, we have added some example of environmental factor and habits.
Page 8, Lines 294-296: “Parents, especially mothers, have a strong influence in the home environment and the mother’s education level can affect certain environmental factors (e.g. TV in the child’s bedroom) and habits (e.g. background TV) that influence RST in the child”.
